# Pilot Double-Blind Randomised Controlled Trial: Effects of Jejunal Nutrition on Postprandial Distress in Diabetic Gastropathy (J4G Trial)

**DOI:** 10.3390/nu14071321

**Published:** 2022-03-22

**Authors:** Lucianno Carneiro, Jonathan White, Helen Parker, Caroline Hoad, Emily Tucker, Luca Marciani, Penny Gowland, Tasso Gazis, Marjorie Walker, Mark Fox

**Affiliations:** 1NIHR Nottingham BRC, Nottingham University Hospitals NHS Trust and the University of Nottingham, Nottingham NG7 2UH, UK; lucianno.carneiro@outlook.com (L.C.); jonathan.white@nottingham.ac.uk (J.W.); helen.parker38@nhs.net (H.P.); ppzclh@exmail.nottingham.ac.uk (C.H.); emilytucker@doctors.net.uk (E.T.); mszlm1@exmail.nottingham.ac.uk (L.M.); ppzpag@exmail.nottingham.ac.uk (P.G.); tasso.gazis@nuh.nhs.uk (T.G.); 2Department of Gastroenterology and Hepatology, University Hospital Zürich, CH-8091 Zürich, Switzerland; 3Digestive Function: Basel, Center for integrative Gastroenterology, Klinik Arlesheim, CH-4144 Arlesheim, Switzerland; 4Sir Peter Mansfield Imaging Centre, School of Physics and Astronomy, University of Nottingham, Nottingham NG7 2RD, UK; 5School of Medicine & Public Health, Faculty of Health and Medicine, University of Newcastle, Callaghan, NSW 2308, Australia; marjorie.walker@newcastle.edu.au

**Keywords:** diabetic gastropathy, gastroparesis, gastric emptying, MRI, jejunal nutrition

## Abstract

Nausea, vomiting and abdominal pain in diabetic patients are often attributed to diabetic gastropathy (DG). Post-pyloric (“jejunal”) enteral nutrition (JN) may improve nutrition and glycaemia in difficult cases. The acute effects of JN on postprandial symptoms and gastric function in DG patients has not been studied. DG patients with moderate to severe symptoms (gastroparesis cardinal symptom index (GCSI) > 27), diabetic controls without symptoms (DC; GCSI < 14) and healthy controls (HV) were entered into a randomized, double blind controlled trial. JN with liquid nutrient (2 kcal/min) or water was infused for 60 min prior to ingestion of a standardized mixed solid/liquid test meal. Outcomes included postprandial symptoms and effects on gastrointestinal (GI)–peptide hormones and gastric emptying (GE) assessed by magnetic resonance imaging (MRI). Nine DG, nine DC and twelve HV were recruited. DG patients reported more symptoms after meals than other groups (*p* < 0.05). Post-prandial symptoms were reduced after JN in DG patients (*p* < 0.01). GE was more rapid after JN in DG and DC patients (*p* < 0.05). JN induced a GI–peptide response in all subjects; however, this was less pronounced in diabetic groups. JN has beneficial effects on DG patients’ symptoms after a meal. The mechanism is not primarily mediated by effects on GE, but appears to involve other aspects of GI function, including visceral sensitivity.

## 1. Introduction

Diabetic gastroparesis is characterized by upper gastrointestinal symptoms and delayed GE without evidence of mechanical obstruction [1]. A recent systematic review identified an association between measurements of delayed gastric emptying (GE) and upper gastrointestinal symptoms [2]. However, correlation between postprandial symptoms and objective measurements of GE is weak, and a recent study from the NIH Gastroparesis Clinical Research Consortium indicates that functional dyspepsia and gastroparesis are interchangeable syndromes with common clinical and pathologic features [3]. For these reasons, the term diabetic gastropathy (DG) is preferred to gastroparesis in this study. The management of DG is challenging. Patients do not always respond to dietary modification, tight glycaemic control, or standard medications [1]. Endoscopic and surgical therapies have shown inconsistent clinical improvements in clinical trials, [4,5] thus highlighting the need for safe and effective new treatments.

Ideally, treatment would target the underlying pathophysiological basis of symptoms and disease. Physiological investigations in DG patients have shown an abnormal response to oral nutrition on multiple levels, including impaired gastric relaxation, reduced antral contractility and spasm of the pyloric sphincter in addition to abnormal GE [6,7]. Additionally, a blunted gastrointestinal (GI)–peptide hormonal response to enteral feeding (e.g., decreased gastrin and incretin hormones, elevated glucagon, failure to suppress ghrelin) has been reported [8,9].

Supplementary jejunal nutrition (JN) has long been used in the management of DG patients who are unable to meet nutritional requirements. Symptom improvement in this group is thought to follow long-term improvements in diabetic control and nutritional status. However, in our clinical practice, we observed that DG patients hospitalized with severe symptoms can experience prompt relief during and for a few hours after JN. These observations are supported by case reports [10], implying that it provides more than a simple, supportive effect. Based on these observations, a prospective, cross-over study was designed to test the hypothesis that delivery of JN prior to a test meal will reduce postprandial distress in DG patients.

The primary aim of this pilot study was to identify the short-term effects of this novel dietary intervention on postprandial GI symptoms in DG patients, diabetic controls without symptoms (DC) and healthy volunteers (HV). Secondary aims were to assess the effects of JN on gastric function (e.g., contractile wave frequency, gastric emptying) and to identify neuro-endocrine processes that have effects on gastrointestinal motility or sensations.

## 2. Materials and Methods

### 2.1. Participants

Adult patients (18–55 years) with DG and DC attending gastroenterology and endocrinology clinics at Nottingham University Hospital NHS Trust were recruited between 1 April 2013–2 February 2015. HV were recruited by advertisement within the University. The protocol was approved by the NRES Committee East Midlands (REC Ref: 12/EM/0013) and was prospectively registered at ClinicalTrials.gov (NCT00944593, NCT01919021). Written informed consent was obtained.

DG Patients were included if they had the diagnosis of type I diabetes mellitus confirmed by low C-peptide level and history of persistent insulin treatment from diagnosis with at least moderate symptoms (gastroparesis cardinal symptom index; GCSI > 27) [11]. DC and HV had, at most, mild symptoms (GCSI < 14). Exclusion criteria included significant co-morbidities (active or severe cardiovascular, respiratory, or neurological diseases, decompensated liver disease (i.e., cirrhosis or hepatitis)), history of GI disease (inflammatory bowel disease, GI strictures, active peptic ulcer disease) or surgery, pregnancy, poorly controlled glycaemia (HbA1C > 12%), patients taking medications that influence gastric motility (metoclopramide, antibiotics), contra-indications to magnetic resonance imaging (MRI), body mass index (BMI) <18 or > 35, weight > 120 kg, and waist circumference > 99 cm.

### 2.2. Study Screening

At the screening visit, participants underwent a detailed medical history and physical examination with anthropometric measurements, cardiovascular monitoring, urine dipstick and routine blood tests. Questionnaires were completed to assess symptom severity (GCSI) and psychometric status (Perceived Health Questionnaire (PHQ15), Hospital Anxiety and Depression Score (HADS)). The participants then completed a validated 400 mL (300 kcal) Nutrient Drink Test over 10 min to ensure that all could tolerate the study intervention and to screen for the occurrence of postprandial symptoms. This abbreviated drink test is limited to 400 mL of a 0.75 kcal/mL liquid nutrient [12]. This is more practical for application in clinical and research studies than drinking tests that continue until full satiation. Participants recorded dyspeptic symptoms using a validated 100 mm visual analogue scale (VAS) [12]. Symptom definitions followed the recommendations by Revicki et al. [13]. Intake was stopped if severe symptoms occurred (VAS > 90 mm). For reference, on completion of the meal, more than 90% of patients with functional dyspepsia but no healthy volunteers report at least moderate fullness and more than mild dyspeptic symptoms.

## 3. Randomisation and Study Procedures

Patients were randomised in the ratio 1:1 to receive either the test intervention (liquid nutrient) or placebo (water) based on computer-generated random permuted blocks of randomly varying size, created by the Nottingham Clinical Trials Unit. Treatment allocations were blinded to all study personnel and patients. Test fluids were prepared in the hospital pharmacy and concealed by an opaque bag and infusion set. Figure 1 provides an overview of study procedures. Participants were studied on two occasions (cross-over design) no more than 10 weeks apart. Blood sugar monitoring and insulin infusion rate were managed using the Insulin-Sliding Scale (50 units of Actrapid^®^ to 49.5 mL of saline) with an insulin perfusion range between 0.5–5 units/h according to current blood glycemia, aiming for a value between 5–8 mmol/L.

### 3.1. Endoscopy and Histology

Gastroscopies were performed under conscious sedation. Biopsies were obtained from the duodenum and stomach. An NJ feeding tube (2.6 mmÆ, 120 cm, Freka^®^ Tube) was then placed into the jejunum under direct vision. 

Biopsies were processed and stained with haematoxylin and eosin. Gastric biopsies were scored as per the Sydney system, duodenal biopsies were assessed for active inflammation, architecture and intraepithelial lymphocyte count (IEL)/100 enterocytes. Additionally, eosinophils/mm^2^ and immunostaining (CD117) for mast cells were performed. 

### 3.2. Gastric Magnetic Resonance Imaging 

Participants were scanned using a 1.5 T whole-body MRI system (Achieva, Philips, Best, The Netherlands). Baseline imaging measured fasting gastric volume. The study intervention (200 mL JN at 2 kcal/min or 200 mL water over 50 min) was delivered with blinding maintained by an opaque delivery set. Then, the mixed Nottingham Test Meal (NTM) was ingested. This included 400 mL Fortisip Vanilla (Nutricia Clinical; diluted 1:1 with water to 0.75 kcal/mL, 300 kcal) labelled with paramagnetic contrast (0.5 mmol/L Gd-DOTA; Dotarem^®^, Laboratorie Guerbet, Aulnay-sous-Bois, France). Additionally, 12 agar beads (10–12 mm diameter) with known breaking strength (0.8 N/m^2^) were ingested to assess solid gastric emptying [12,14,15]. Gastric filling was documented after 200 mL and 400 mL ingestion (rate of 40 mL and 1–2 agar beads/min (10 min total)). Subsequently, total gastric and gastric meal volumes were recorded at regular intervals. Prior to the final image, patients drank 200 mL of water to facilitate visualization of agar beads in the stomach. MRI acquisition followed published protocols [12]. Gastric motility scans were obtained from 6–8 transverse oblique images covering the luminal wall using a dynamic bFFE sequence with parallel imaging sensitivity encoding. 

### 3.3. Symptom Assessment

A validated, self-report VAS was used to assess GI symptoms during ingestion of the standardized 400 mL liquid nutrient meal and subsequent emptying (0 = none; 10 = severe) [12]. Fullness, satiety, nausea, heartburn, epigastric pain and bloating were documented before each volume measurement.

### 3.4. Image Analysis

Total gastric volume (TGV) was outlined in each image of an MRI volume scan by manually placing reference points on the stomach wall and fitting a closed contour through these points. Gastric content volume (GCV) was identified by applying a manually selected intensity threshold. Summing the identified pixels with intensity larger than the selected intensity threshold and integrating the sum over all slices resulted in the meal volume [16].

A characteristic pattern of volume change is observed after ingestion of the liquid nutrient meal in patients and controls. As described previously [12], for liquid nutrient, “early phase” GE begins during meal ingestion and is driven by mechanical factors (e.g., meal volume) and not by nutrient feedback. This is assessed by measurement of gastric volumes at time = 0 immediately after meal ingestion (TGV0, GCV0). By contrast, “late phase” GE is also modulated by chemical factors (e.g., calorie density). This is assessed by half time (T50) and the rate of GE (mL/min) at T50.

### 3.5. GI–Peptide Hormone Assay

Blood samples were taken at specific time intervals before NJ enteral infusion and then after meal ingestion. Plasma was stored at −28 °C until processed according to manufacturer’s instructions. Processed GI–peptides included insulin (IMx Insulin, Abbott Laboratories, Wiesbaden, Germany), C-peptide (DRG Instruments GmbH, Marburg, Germany), GLP-1 and PP hormones (details of analytic kits in Appendix A).

### 3.6. Statistical Analysis 

Pilot trials are exploratory studies limited in size that are designed to give insight into the actions, efficacy, and safety of an intervention and to provide data that will inform future studies that can deliver definitive support for specific mechanistic or therapeutic claims [17]. 

The primary outcome of this pilot study was patient symptoms after a validated test meal [15]. This was preferred to measurements of gastric function because the effects of prior JN on gastric function were uncertain. Only patients with at least moderate symptom severity were recruited to facilitate detection of a clinically relevant treatment effect. Guidelines for treatment studies in patients with functional GI disease recommend allowing for 40% day-to-day variability of symptoms [18]. Based on these conservative assumptions, power calculations indicate that 12 participants in each group would provide an 80% chance of detecting a clinically relevant (>40%) effect on postprandial symptoms by the study intervention (alpha < 0.05) and that nine participants in each group would detect a trend sufficient to inform design of a definitive trial (alpha < 0.10).

Data were stored in a Microsoft Access database and analysed by a professional statistician (menne-biomed.de) with R [19] and package *knitr* [20]. Mixed model analysis compared the response to intervention and between the three groups. Data with normal residual distribution were processed by linear mixed models; package *lme* [21] results were reported as estimates, 95% confidence intervals (CI) and *p* values. Bayesian methods with package *rstanarm* [22] were used for data processing. Bayes estimation gives credible intervals (Crl) with 95% confidence. This model determines the effects of external factors on GE parameters. Non-linear Bayesian fits of gastric volume time series with LinExp parametrization were computed with Stan [23] and strong gamma priors to the range of 0.2–2 for parameter kappa. Demographic data are reported as median and quartiles of clinical scores. Non-parametric Kruskal-Wallis tests were used for group comparisons. For continuous data analysis, Wilcoxon test was used. *p* values < 0.05 were statistically significant.

## 4. Results

### 4.1. Study Participants

Thirty-five participants were screened. Two HVs, two DG and one DC patient declined participation. Thirty patients completed the study: twelve HVs, nine DG and nine DC. There was no difference in the demographic features (Table 1) or duration of diabetes, (DG 17 years (interquartile range (IQR): 13.0, 31.0) vs. DC 17 years (IQR: 12.0, 22.0); *p* = 0.86). DG patients had a median of 2 years (IQR: 2.0, 5.0) duration of symptoms. All participants tolerated jejunal tube placement, and there were no complications or reports of poor acceptance that required termination of the investigation during study procedures.

### 4.2. Endoscopic and Histological Findings

There were no significant differences in the histological parameters examined including gastritis, duodenal eosinophil counts or mast cell counts when comparing HV, DG and DC. IEL counts were not significantly higher in DG (DG 23 vs. DC 16; *p* = 0.61).

### 4.3. Effect of JN on Postprandial Symptoms

At screening, DG patients reported higher satiety (*p* = 0.033), bloating (*p* = 0.05) and pain (*p* = 0.05) after ingestion of the 400 mL Nutrient Drink Test when compared with DC and HV (Figure 2). During the interventional studies, when JN was delivered before the oral test meal, postprandial fullness, bloating and pain were reduced in DG patients. The effect size was modest (typically reduction in VAS 10–20) but consistent (*p* < 0.01 for all symptoms, except nausea). By contrast, the nutritional intervention did not influence postprandial symptoms in DC or HV (Figure 3).

### 4.4. Gastric Volume, Motility and Emptying

In all participants, GE commenced during ingestion of the liquid NTM as evidenced by the presence of Gadolinium-labelled liquid in the small bowel in the first image acquired after meal ingestion (Figure 4). “Early-phase” GE as assessed by GCV0 immediately after the oral meal was higher in all three groups if JN rather than water (placebo) had been delivered prior to the test meal (e.g., HV; +31 mL (StdErr 12.6 mL), *p* = 0.019) with similar results for TGV0 (e.g., HV; +42 mL (StdErr 14.1 mL), *p* = 0.0056). This “early-phase” emptying was followed by a “late-phase” linear–exponential reduction in meal volume. “Late phase” GE as assessed by half time (T50) and the rate of GE at T50 was similar following JN or water in all three groups (*p* = 0.727). Note that four DG patients, two DC and no HV had abnormally slow liquid GE during the placebo phase compared to published reference intervals for the NTM [12]. The response of these individuals to the nutritional intervention was qualitatively similar to other patients.

Solid GE, defined by the number of intact agar beads (12 total) that had emptied from the stomach, was similar in HV and both patient groups. However, solid emptying was more rapid if JN had been delivered prior to the NTM in both DC and DG patients (+2 (IQR 1–3), vs. +3 (IQR 1–7); both *p* < 0.05, compared to placebo).

Antral contraction wave frequency was 2.7/min in HV and was numerically higher in both patient groups (DC (3.1/min), DG (2.9/min)), (*p* = 0.57). This measurement was not significantly affected by the intervention. 

### 4.5. GI Neuroendocrine Response

Infusion of liquid nutrition but not water into the jejunum induced a GLP-1 and PP response in all groups (Figure 5). This effect was less pronounced in both patient groups than in HVs. Compared to HVs, the increase in GLP-1 with JN was less marked in DC −0.92 (CrI −2.1 to 0.27) pmol/L and DG patients −0.38 (CrI −1.6 to 0.73) pmol/L with similar effects observed for PP concentration. There was no correlation between measurements of GI–peptides with symptoms or GE.

## 5. Discussion

This pilot trial applied a prospective, randomized, placebo-controlled study design to assess the effects of jejunal nutrition (JN) delivered before a test meal on postprandial symptoms and gastric function in patients with diabetic gastropathy (DG), diabetic controls and healthy volunteers. Beneficial effects of this intervention on fullness, bloating and pain were observed in the DG group. This novel dietary intervention had consistent effects on early gastric emptying and on the release of GI–peptide hormones; however, it was not associated with *specific* changes in DG patients. Thus, the mechanism by which JN reduces postprandial symptoms is uncertain but could be mediated by effects on gastric accommodation that were present in all groups and/or effects on gastric sensory function in DG patients. 

The results of this study are consistent with the primary hypothesis that JN delivered before ingestion of an oral test meal can improve postprandial symptoms in DG patients. The potential clinical relevance of this finding is supported by case reports of rapid, quasi-pharmacological improvement of individuals that received JN as part of nutritional management of DG [10]. Age- and sex-matched disease controls were recruited to ensure that findings were not related to the underlying disease state, but were specific to DG patients. As expected, higher scores for anxiety and depression and lower quality of life were recorded by the symptomatic group, underlining the impact of this condition on quality of life and the need for new approaches to treatment. 

Detailed measurement by MRI of gastric function and assay of GI–peptide hormones did not identify a specific physiological mechanism that explains the effects of JN on postprandial symptoms in DG patients. The nutritional intervention was not associated with changes in antral contraction wave frequency, gastric volume (TGV) during meal ingestion, or gastric emptying that were only seen in the symptomatic group. This was not because the “dose” of JN administered prior to the oral test meal was too small. The 200-kcal infusion was sufficient to trigger the release of GI–peptide hormones, relax the stomach (i.e., promote accommodation as assessed by TGV) and slow “early-phase” GE in healthy volunteers and diabetic patients with and without symptoms. This is good evidence that JN activated neurohormonal feedback and the “small bowel brake” [24]. Consistent with the study hypothesis, these findings suggest that JN relaxed the stomach in all participants and that this reduction in wall tone/tension reduced postprandial distress in symptomatic patients. The failure to identify a specific mechanism by which JN improves symptoms in DG patients is likely because of two factors. First, diabetes has heterogeneous effects on gastric function, ranging from a major motility disorder in which symptoms are directly related to abnormal GE (“true gastroparesis”) to abnormalities such as impaired gastric relaxation (“accommodation”), and visceral hypersensitivity that are also present in patients with functional dyspepsia [25,26,27,28]. Second, there are limitations of non-invasive imaging in the assessment of gastric function. Barostat studies have shown that JN promotes gastric accommodation in healthy controls and, to a lesser extent, patients with functional dyspepsia [29]. It was not possible to demonstrate similar findings in DG patients in this study because non-invasive imaging has sub-optimal sensitivity to changes in gastric tone, unless combined with intra-luminal pressure measurement [30]. Notwithstanding these issues, whether the effect of JN was to promote accommodation or to reduce visceral sensitivity to gastric distension, DG patients reported significantly reduced abdominal fullness, bloating and pain if JN was delivered before ingestion of the oral test meal.

An excess number of mucosal inflammatory cells in the duodenal mucosa have been noted in subjects with autoimmune conditions and functional dyspepsia [31]. This was not confirmed by histology in this cohort. This may be due to a lack of study power for this secondary outcome, or it could be that DG has a different aetiology.

This study included a relatively small number of participants and controls; however, significant results were obtained, and the numbers were sufficient for the purposes of a pilot investigation. Another limitation is that no single investigation can fully assess the GI response to feeding. MRI measurements during the inter-digestive, fasted phase are difficult because imaging requires the presence of luminal contents to visualize gastric and small bowel contractions. In addition, as discussed above, assessment of gastric accommodation is challenging because it is the size of the meal and not gastric wall compliance that is the main determinant of volume change during meal ingestion [30].

In conclusion, this clinical physiological study provides initial evidence that the delivery of JN prior to ingestion of an oral test meal can reduce postprandial symptoms in DG patients. If confirmed, then JN could provide a novel, non-medical management tool for DG patients that have not responded to supportive care with antiemetics or promotility agents, a situation in which the lack of effective treatment is widely acknowledged. The results also suggest that a better understanding of the GI response to feeding and the underlying pathophysiology of postprandial symptoms in DG patients could help optimize nutritional interventions in this difficult-to-treat condition. It may also help to identify novel pharmacological targets for therapy in DG and, potentially, related conditions such as idiopathic gastroparesis and functional dyspepsia.

## Figures and Tables

**Figure 1 nutrients-14-01321-f001:**
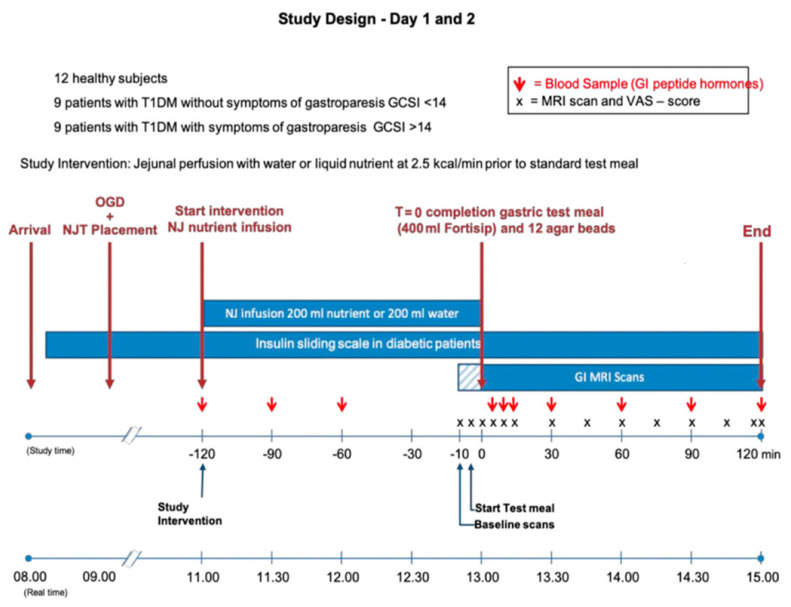
Study design with flow diagram detailing patient progress and procedures during the study. The NJT was placed under sedation during endoscopy. After 2 h recovery, the study intervention was delivered over 120 min. The oral meal was ingested over 10 min, and study measurements were acquired. NJT, naso-jejunal tube; M, meal; MRI, magnetic resonance imaging; OGD, Oesophago-Gastro-Duodenoscopy; GI, gastrointestinal; VAS, visual analogue scale, GCSI, gastroparesis cardinal symptom index.

**Figure 2 nutrients-14-01321-f002:**
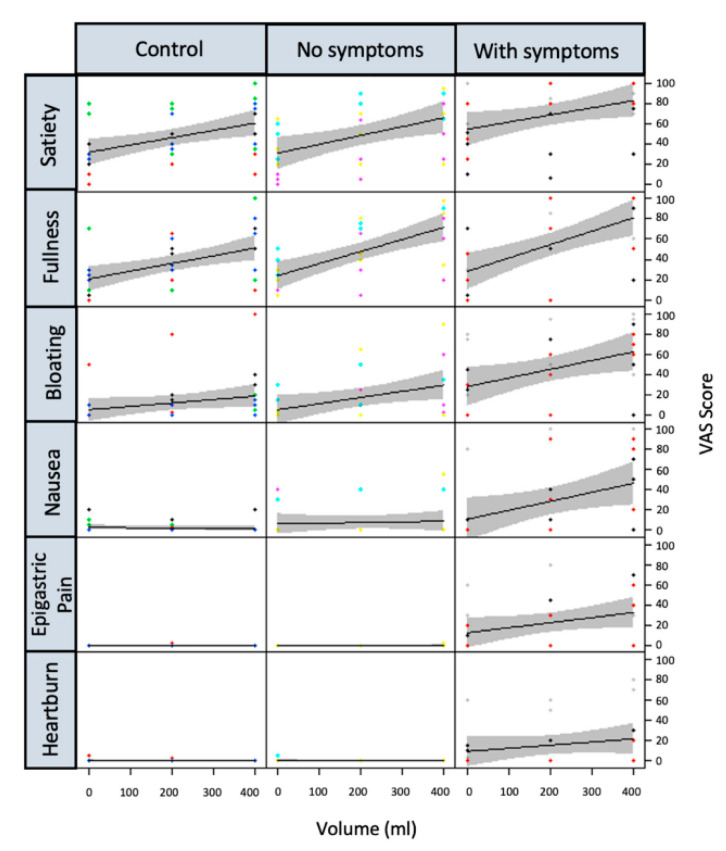
Postprandial symptoms recorded on visual analogue scale (100 mm VAS) during oral ingestion of the liquid Nottingham Test Meal (NTM) with 95% confidence bands at screening. Symptoms were recorded at baseline and after ingestion of 200 and 400 mL of the test meal. Satiety and fullness were similar in all three groups; however, dyspeptic symptoms were more marked in diabetic gastropathy (DG) patients. “Control”, healthy volunteers; “no symptoms”, diabetic controls without gastroparesis; “with symptoms”, patients with symptoms characteristic of diabetic gastroparesis.

**Figure 3 nutrients-14-01321-f003:**
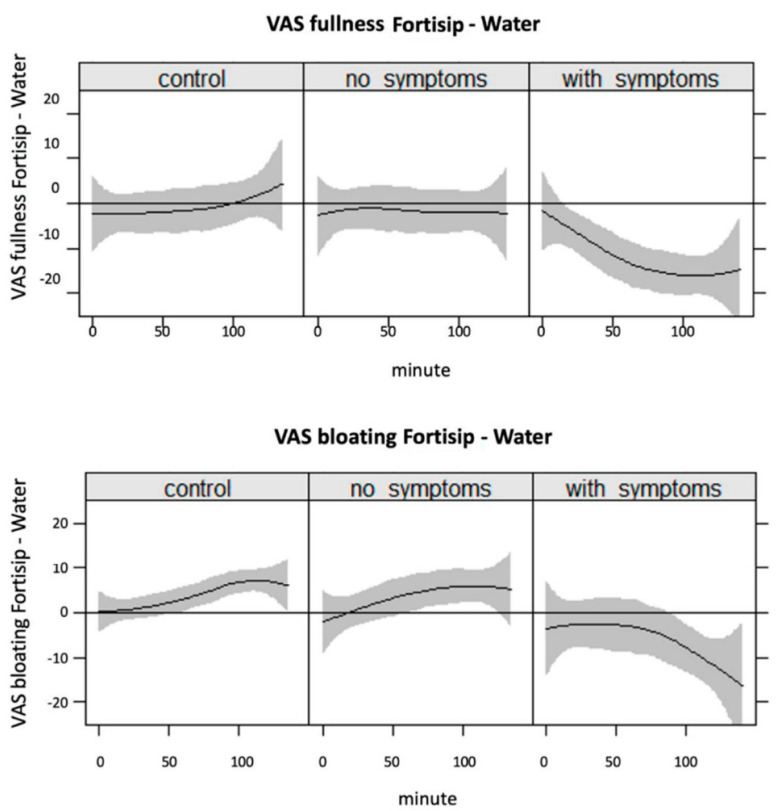
Variation in symptoms between individuals is large. For this reason, this figure presents the difference in postprandial symptom severity following ingestion of the oral test meal after jejunal infusion with liquid nutrient (JN) and placebo (water) with 95% confidence bands. Negative values represent a reduction in symptoms after the nutritional intervention. Healthy controls (HV) and diabetic controls without symptoms (DC) reported similar or non-significantly higher postprandial symptom scores if JN had been delivered prior to the meal. In contrast, DG patients had significantly less postprandial fullness, bloating and pain.

**Figure 4 nutrients-14-01321-f004:**
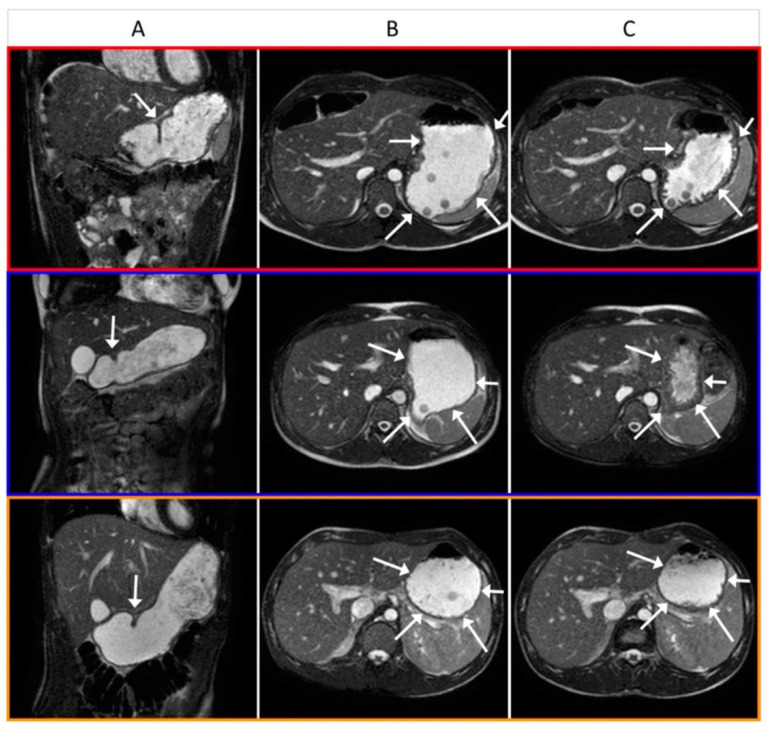
Magnetic resonance (MR) images of the stomach from representative participants. Red outlined images from an HV, blue from a DC, and orange from a DG patient. Column A: A coronal slice through the stomach taken 15 min after ingestion; white arrow shows antral contraction in wall. Column B: An axial slice through the stomach taken immediately after ingestion; white arrows highlight the thin “distended” stomach wall. Column C: An axial slice through the stomach taken 90 min after ingestion; white arrows highlight the thicker stomach wall for the red and blue images after much of the meal has emptied in these subjects, whilst the orange image shows little change in volume or wall thickness compared to the B image, as little has emptied from the stomach.

**Figure 5 nutrients-14-01321-f005:**
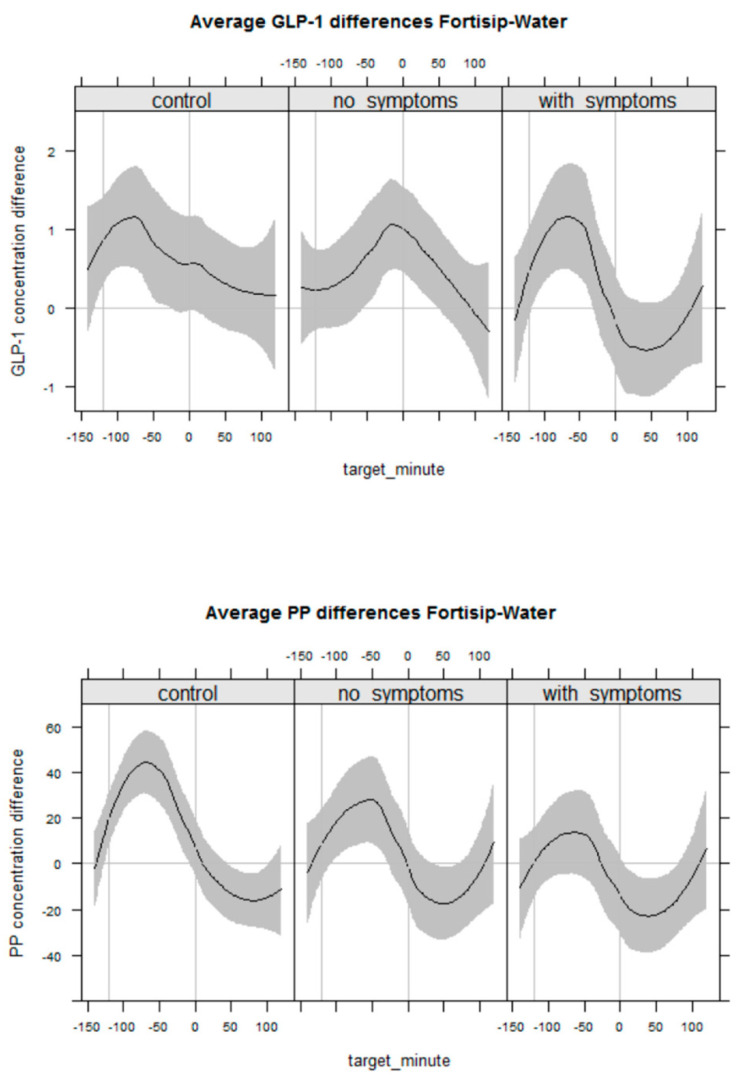
Variation in GI–peptide hormone levels between individuals is large. For this reason, this figure presents the change in mean GLP-1 (upper panel) and PP concentrations (lower panel) during delivery of jejunal nutrition (Fortisip) and water (−120 to 0 min.) and after ingestion of the oral test meal (0 to +120 min.), by participant group, with confidence bands before and after jejunal infusion. Positive values indicate greater GLP-1 and PP after the nutritional intervention. GLP-1, glucagon-like peptide-1; PP, pancreatic polypeptide.

**Table 1 nutrients-14-01321-t001:** Demographic and participants characteristics.

	HV (GCSI < 14)	DC (GCSI < 14)	DG (GCSI > 27)	*p* Value
Number	12	9	9	-
Male sex (Percentages)	5 (41.7)	3 (33.3)	5 (55.6)	0.629
BMI	26.70 (22.5–30.18)	22.20 (21.5–25.1)	22.40 (20.4–25.2)	0.213
Age (years, range)	30.50 (24.75–41.25)	32.00 (29–38)	30.00 (27–34)	0.724
Anxiety score (range)	2.50 (1–3.25)	3.00 (1–4)	5.00 (4–10)	0.013
Depression score	0 (0–0.25)	0 (0–1)	4 (4–10)	0.003
PHQ 15 Score	1.50 (0–2)	2.00 (1–3)	10.00 (7–13)	<0.001
PHQ15 quantile	37.40 (0.38–46.97)	38.60 (23.9–58.40)	96.70 (92.6–97.60)	0.001
Smoking				0.156
Current smoker (percentage)	4 (33.3)	2 (22.2)	4 (44.4)	
Ex-smoker (percentage)	1 (8.3)	2 (22.2)	4 (44.4)	
Never smoked (percentage)	7 (58.3)	5 (55.6)	1 (11.1)	
Alcohol				0.189
<20 units per week (percentage)	9 (75.0)	9 (100.0)	5 (55.5)	
Never (percentage)	3 (25.0)	0 (0.0)	4 (44.4)	

Median and quartiles for clinical scores. Non-parametric Kruskal-Wallis tests were used for group comparisons. Hospital Anxiety and Depression Score (HADS) scales used for anxiety and depression scores. Quantile Perceived Health Questionnaire (PHQ-15) scores are corrected for age and gender. HV, healthy controls; DC, diabetic controls without symptoms; DG, diabetic gastropathy; BMI, body mass index.

## Data Availability

The data are available on request.

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
