# Peer review of "Pilot Double-Blind Randomised Controlled Trial: Effects of Jejunal Nutrition on Postprandial Distress in Diabetic Gastropathy (J4G Trial)"

_nutrients, 2022, doi:10.3390/nu14071321_

Round 1
Reviewer 1 Report
This is a very well designed and presented study, with possible clinical implications. However I have some questions regarding methods and result presentation:
- What was the acceptance and tolerance of jejunal tube in study participants?
- Where there any complications of jejunal feeding other than mentioned in the study results (e.g. diarrhoea)
- Where there any differences in jejunal feeding acceptance between study meal and water?
- Is it safe to infuse hypotonic and hyposmolaric fluids into jejunum? - please comment
Author Response
Reviewer #1
This is a very well designed and presented study, with possible clinical implications. However I have some questions regarding methods and result presentation:
- What was the acceptance and tolerance of jejunal tube in study participants?
We are grateful for the positive comments of the reviewer.
The naso-jejunal feeding tubes were placed under sedation during endoscopy (see method and now added to legend). All patients tolerated the jejunal tube placement and there were no complications or reports of poor acceptance / need for removal during the study time period. Sentence added to the results. “All participants tolerated jejunal tube placement, there were no reports of poor acceptance or complications during the study period.”
- Were there any complications of jejunal feeding other than mentioned in the study results (e.g. diarrhoea)
There were no reported complications or need to deviate from the protocol to go the toilet or for other reasons during the duration of the test.
- Where there any differences in jejunal feeding acceptance between study meal and water?
Abdominal symptoms were the primary outcome of the study. Diabetic patients with gastropathy had less abdominal pain and other postprandial symptoms following jejunal feeding. For the healthy and disease controls there were no significant differences between the two study arms.
- Is it safe to infuse hypotonic and hyposmolaric fluids into jejunum? - please comment
There is no indication from extensive clinical and research experience that infusion of water or nutrients directly into the jejunum at the (standard) rates used in this study is risky.
Reviewer 2 Report
Authors have evaluated the effect of JN on the post prandial symptoms in patients with DG comparing with DC, HV using a Randomized pilot study. I do have few queries.
Methods: Please report/elaborate what significant comorbidities, GI diseases are.
Why HbA1c>12% was excluded?
Please elaborate on the flow of events in the methods. It is hard to follow.
It is unclear from the figures to interpret what is the difference between JN and water? Kindly elaborate in the figure what you are demonstrating and the significance value.
What is the clinical relevance of this study? Do authors recommend post pyloric feeding for DG or to trial promotility agents which would be easily doable rather than opting for post pyloric feeding.
Author Response
Authors have evaluated the effect of JN on the post prandial symptoms in patients with DG comparing with DC, HV using a Randomized pilot study. I do have few queries.
- Methods: Please report/elaborate what significant comorbidities, GI diseases are.
Exclusion criteria included active or severe cardiovascular, respiratory or neurological disease, decompensated liver disease (i.e., evidence of cirrhosis or hepatitis), inflammatory bowel disease, GI strictures and active peptic ulcer disease were excluded as these disorders could either prevent safe gastroscopy and jejunal tube placement or impact on motility / digestion. More details are added to method.
- Why HbA1c>12% was excluded?
It is known that hyperglycaemia can slow gastric emptying and that changes in blood glucose affect appetite and gastric function (e.g., Andrews, Rayner, Doran, Hebbard and Horowitz, Am J Physiol 1998). Additionally, in order to obtain homogeneous study groups, it was considered important to exclude poorly controlled diabetes.
- Please elaborate on the flow of events in the methods. It is hard to follow.
The methods and the figure 1 legend have been edited to improve clarity
- It is unclear from the figures to interpret what is the difference between JN and water? Kindly elaborate in the figure what you are demonstrating and the significance value.
Variation in symptoms between individuals is large. For this reason, Figure 3 presents the difference in symptom severity between the two study conditions is presented. Negative values demonstrate a reduction in symptoms after the nutritional intervention. The severity of postprandial bloating and fullness reported by diabetic patients with gastroparesis is less after prior jejunal feeding than after jejunal water. The legend of figure 3 has been edited to improve clarity.
Note that, for the same reasons the legend of figure 5 has also been edited.
- What is the clinical relevance of this study?
This is a pilot study and the results must be confirmed by a larger clinical study; however, the findings may have clinical relevance. “Jejunal feeding represents a novel, non-medical management tool for diabetic gastroparesis, a condition for which the lack of effective treatment is widely acknowledged.” Sentence added to conclusion.
- Do authors recommend post pyloric feeding for DG or to trial promotility agents which would be easily doable rather than opting for post pyloric feeding.
At present a trial of promotility agents should be first line due to costs and limited access to nutritional specialist services. Going forwards, improved understanding of the underlying pathophysiology of postprandial symptoms in patients with diabetic gastropathy (and perhaps related conditions, such as functional dyspepsia) may identify additional targets for pharmacological treatment. Sentence added to conclusion.
Reviewer 3 Report
1- The title has not been selected properly. You should add pilot double-blind in the title.
2- Your study details (study date, inclusion/exclusion criteria and …) are different in the registry system and your manuscript.
3- Why did you postpone your report for about 6 years?!
Author Response
We are grateful for the input of this reviewer and the opportunity to respond to these important points.
- The title has not been selected properly. You should add pilot double-blind in the title.
Title amended. “Pilot Double-Blind Randomised Controlled Trial: Effects of Jejunal Nutrition on Postprandial Distress in Diabetic Gastropathy (J4G Trial)”
- Your study details (study date, inclusion/exclusion criteria and …) are different in the registry system and your manuscript.
Originally the trial was registered in Zürich. The lead author changed institutions and the funding followed the researcher to the University of Nottingham. Due to a bureaucratic error, the registry entry was canceled and not updated. The correct study details as stated in the manuscript were specified in the ethics submission submitted and approved (University of Nottingham REC Ref: 12/EM/0013) prior to study commencement.
- Why did you postpone your report for about 6 years?!
Delay in publication was due to multiple unavoidable factors. Team members moved institution, there were delays in assessment of GI peptide hormones and tissue processing, and comments from reviewers required re-analysis and re-presentation of the data.
Notwithstanding these issues, we hope that the reviewers agree that the decision to proceed with (delayed) publication was considered preferable to not disseminating this novel and potentially clinically relevant data.